# Pulse Wave Velocity, a Predictor of Major Adverse Cardiovascular Events, and Its Correlation with the General Stress Level of Health Care Workers during the COVID-19 Pandemic

**DOI:** 10.3390/medicina58060704

**Published:** 2022-05-26

**Authors:** Ioana Marin, Mircea Iurciuc, Florina Georgeta Popescu, Stela Iurciuc, Calin Marius Popoiu, Catalin Nicolae Marin, Sorin Ursoniu, Corneluta Fira-Mladinescu

**Affiliations:** 1Discipline of Occupational Medicine, Department of Internal Medicine V, University of Medicine and Pharmacy “Victor Babeş”, Eftimie Murgu Square, No.2, 300041 Timisoara, Romania; ioana.marin18@gmail.com (I.M.); gflorinag@yahoo.com (F.G.P.); 2Discipline of Internal Medicine and Ambulatory Care, Prevention and Cardiovascular Recovery, Department of Cardiology, University of Medicine and Pharmacy “Victor Babes”, Eftimie Murgu Square, No.2, 300041 Timisoara, Romania; mirceaiurciuc@gmail.com (M.I.); stela_iurciuc@yahoo.com (S.I.); 3Department of Pediatrics, University of Medicine and Pharmacy “Victor Babes”, Dr. Iosif Nemoianu Street, No.2, 300011 Timisoara, Romania; mcpopoiu@yahoo.com; 4Department of Physics, Faculty of Physics, West University of Timisoara, V. Parvan Ave., No.4, 300223 Timisoara, Romania; catalin.marin@e-uvt.ro; 5Discipline of Public Health, Department of Functional Sciences, Center for Translational Research and Systems Medicine, University of Medicine and Pharmacy “Victor Babes”, Eftimie Murgu Square, No.2, 300041 Timisoara, Romania; 6Discipline of Hygiene, Department of Microbiology, Preventive Health Education Center, University of Medicine and Pharmacy “Victor Babes”, Eftimie Murgu Square, No.2, 300041 Timisoara, Romania; fira-mladinescu.corneluta@umft.ro

**Keywords:** COVID-19, health care workers, general stress, pulse wave velocity

## Abstract

*Background and Objectives:* In the COVID-19 epidemiological context, the health care workers who were treating patients with COVID-19 were exposed daily to additional stress. Pulse wave velocity (PWV) is a predictive parameter for possible major adverse cardiovascular events. The present study aimed to evaluate the correlation between the general stress levels and PWVs of medical workers during the COVID-19 pandemic. *Materials and Methods*: The study group was heterogeneous in terms of the medical profession. PWV was measured using a TendioMed arteriograph. Assessment of stress level was performed using a general stress questionnaire with questions grouped on the areas that contribute to stress: lifestyle, environment, symptoms, job, relationships and personality. PWV measurements and stress assessment were performed both during the period with many patients with COVID-19 and during the period with few patients with COVID-19. *Results*: The stress levels and PWVs of subjects were higher in the period when they cared for patients with COVID-19 than in the period when they did not have patients with COVID-19. *Conclusions*: The study shows a positive correlation between the PWV of each subject and his/her stress score (the higher the stress score, the higher the PWV).

## 1. Introduction

The successive waves of the COVID-19 pandemic have affected countries all over the world and health systems have been overwhelmed by the rapid evolution of the SARS CoV-2 novel coronavirus. Due to the complexity of the pandemic situation, some health care workers suffer from a high degree of stress, and studies have shown the need for psychological support [1,2].

Assessing and understanding the relationship between stress at work and the risk of getting ill from any kind of disease can provide tools for managing stressful situations and avoiding the occurrence of pathological consequences [3,4].

If at the beginning of the pandemic there were acute stress reactions (emotional, cognitive, physical and social reactions, usually present in combination [5]) with the passage of different pandemic waves and prolonging the pandemic more than we would have expected, we can talk about the chronicity of stress at work for health care workers.

Chronic stress is a real health hazard and can lead to diseases or the worsening of existing ones. Some studies show a connection between stress and various liver diseases [6]. There is a correlation between psychosocial stress and the severity of chronic C hepatitis [7]. The same authors found also a positive correlation between psychosocial stress and cirrhosis in their clinical test [7].

Stress hormones secreted in a tumor microenvironment can directly affect tumor cells, promoting their malignant characteristics [8]. The link between stress and stress hormones that particularly sensitize pancreatic tissue was demonstrated [9], so this organ affected by stress hormones can cause diabetes mellitus.

Other studies suggested the link between chronic stress and rapid aging, based on oxidative stress, which was demonstrated to stimulate the loss of telomeres DNA, processes shown to be similar to those occurring during cellular senescence [10].

The stress hormones cortisol and catecholamine regulate blood pressure and blood flow [11,12,13], but their release under stress due to excitation of the sympathetic nervous system causes contraction of the coronary arteries and possibly rupture of vulnerable atheroma plaques at their level [11]. Chronic stress affects lipid metabolism, activates macrophages and promotes the formation of foam cells that lead to the formation of atheroma plaques in the coronary, cerebral and peripheral arteries [11].

Arterial stiffness occurs before atherosclerosis, with the decrease in vascular elasticity being a risk factor for atherosclerosis independent of the classic risk factors. Pulse wave velocity (PWV) is a functional parameter that provides information related to arterial stiffness and the state of ventricular ejection. In fact, it is a predictive parameter of cardiovascular (CV) events, especially in patients with high CV risk. Measuring and analyzing PWV gives information on the potential future development of complications such as atherosclerosis and hypertension, which can lead to major adverse cardiovascular events [12,13,14,15,16,17].

The relationship between psychological stress measured by anxiety and arterial stiffness was explored by Logan et al. [18], who demonstrated that anxiety is a significant and independent determinant of arterial stiffness. On the other hand, Nomura et al. [19] found the association between job stress and arterial stiffness (evaluated by brachial-ankle pulse-wave velocity) to be inconsistent, and authors of another study on evaluating atherosclerotic risk using pulse wave velocity (PWV) in steelworkers employed on different shifts showed no differences in PWV between the shifts [20]. Although there are various studies on the link between mental stress or psychological stress and arterial stiffness, few refer to medical workers.

The present study aimed to evaluate a possible correlation between the general stress levels and pulse wave velocities of medical workers during the COVID-19 pandemic. Medical workers were investigated while they were working with many COVID-19 patients but also at the end of the third pandemic wave (after approximately seven months) when the number of COVID-19 patients was small.

## 2. Materials and Methods

The subjects of the study were employees in two health care departments of the same hospital from Timișoara who cared for COVID-19 patients. These departments were medical units for other pathologies than pulmonary diseases that had been transformed and equipped for the care of COVID-19 patients with mild forms of the disease. The study group consisted of 26 subjects, and it was heterogeneous in terms of the medical profession: medical doctors, resident doctors, nurses and cleaning staff who agreed based on informed consent to participate in the study. The characteristics of the study group are summarized in Table 1.

The subjects in the study were aged between 26 and 58 years and had worked in the profession between 2 and 41 years. Of the total number of subjects, 6 (23.07%) were smokers, and 7 (26.92%) stated that they had cardiovascular diseases (high blood pressure, under antihypertensive treatment). The study subjects had normal work schedules and were not isolated from their families. All the subjects were vaccinated or had recovered from COVID-19.

Pulse wave velocity was measured using a TensioMed arteriograph. The measurements were performed following the recommendations [21]: subjects rested for 10 min in a quiet room; they did not drink alcohol the night before the measurements; and they did not exercise, smoke, or drink coffee for at least 3 h before the measurements. PWV measurements were made in a supine position, with the cuff on the dominant arm, after determining the arm circumference and the length of the distance between the upper curvature of the sternum and the upper edge of the pubis.

The stress level was assessed by applying a specific questionnaire consisting of 96 items and an additional score for some life aspects, as published by Dr. Julian Melgosa [22]. The questions were grouped on the areas that contribute to stress: lifestyle, environment, symptoms, job, relationships and personality. Answers were assessed on a 4-item Likert scale (never, rarely, frequently, and almost always, score from 0 to 3). Five levels of stress were used to interpret the results: level 1: ≤48 points, very low; level 2: 49–72 points, low; level 3: 73–120 points, normal; level 4: 121–144 points, high; level 5: ≥145 points, dangerously high.

Stress levels and arterial stiffness were measured at the beginning of the study when the medical care units had many COVID-19 patients hospitalized and after seven months when only a few COVID-19 patients were hospitalized. The maximum number of patients (in both departments) at the beginning of the study was 52, and the number of patients at the end of the study was 6.

Statistical processing and plotting of the results were performed using ORIGIN 8 software. Means, standard deviations and standard errors of the mean are presented. The paired Student’s *t*-test was used to compare mean values within groups at baseline and at final follow-ups. We performed a power analysis for the paired Student’s *t*-test. Linear regression analysis was performed between PWV and the stress score. The *p* values for all hypothesis tests were two-sided, and we set statistical significance at *p* < 0.05.

## 3. Results

From the analysis of the stress questionnaire scores at the beginning and at the end of the study, we grouped the study subjects into two categories, either normal stress (level 3) or high stress (level 4), as can be seen in Figure 1. The results of the paired-sample *t*-test for the scores of the questionnaire stress are presented in Table 2. One can observe that the null hypothesis is rejected, which means that the stress level of the study group was smaller at the end of the study than at the beginning of the study. The means and standard deviations of the stress questionnaire scores by age group are shown in Table 3, both at the beginning of the study (when the medical workers cared for a large number of COVID-19 patients) and at the end of the study (when the medical workers cared for a small number of COVID-19 patients).

The measurements of PWV versus the age of each subject are presented in Figure 2. The mean PWV at the beginning of the study was 9.12 m/s, and at the end of the study, it was 8.64 m/s. The results of the paired-sample *t*-test for the mean values of PWV are presented in Table 4, showing that the average PWV at the beginning of the study is different from the average PWV at the end of the study. One can observe that the null hypothesis is rejected, which means that the mean value of PWV at the beginning of the study is statistically different from the mean of PWV at the end of the study. The means and standard deviations of PWV by age group are shown in Table 5, both at the beginning of the study (when the medical workers cared for a large number of COVID-19 patients) and at the end of the study (when the medical workers cared a small number of COVID-19 patients).

The dependence of PWV on the stress questionnaire score of the subjects participating in the study is presented in Figure 3, which shows a linear regression line. The results of the linear fit of the dependence of PWV on the stress questionnaire score, as well as the statistical correlation between PWV on the stress questionnaire score, are presented in Table 6.

## 4. Discussion

The assessment of the stress level for the medical workers showed normal and high levels, but no one reached a dangerously high level of stress (see Figure 1). However, when the number of patients with COVID-19 was low, the stress level of the study group was lower than when the number of patients with COVID-19 was high (as can be seen in Figure 1a,b and statistically confirmed by the paired-sample *t*-test presented in Table 2.

An explanation for the fact that the subjects of the study did not reach a dangerously high level of stress may be that they worked at the same job, their work schedule did not change and their teams remained the same. For instance, a study conducted on health care workers in Spain shows that structural changes in the hospital during the first wave of the COVID-19 pandemic affected the mental health of the subjects, and those effects were maintained over time [23].

Another beneficial factor that might have prevented the medical workers in our study from reaching a dangerously high level of stress may be that they were not isolated from their families when they cared for COVID-19 patients (benefiting thus from the family support environment). A study conducted on health care workers in northwest Italy has shown that subjects who decided to live separated from their loved ones experienced more psychological symptoms [24]. Family support in reducing stress was reported in [25]; family showed an extremely powerful influence on individuals. A recent review article [26] also concluded that it is essential to have good communication and positive activities among family members in order to build a sense of togetherness, trust, cohesion and happiness for better responses to crisis and adversity.

There are various studies regarding the perception of stress related to COVID-19 in the general population [27,28,29]. A study conducted on the general population in the United States [29] revealed a higher stress level in the younger population, but in our study, the reverse is found. As can be observed from Table 3, the average stress score of subjects from the 20–24 years old group is smaller than the average stress score of subjects from the 40–60 years old group. This result may be related to the characteristics of COVID-19, which has more severe symptoms in the elder population. Ali-Reza Babapour [30] observed an increase in the stress level of nurses with age, but this was explained as a result of job burnout.

Figure 2 shows the PWV values measured at the beginning of the study (red squares), when the care units had a large number of COVID-19 patients hospitalized and at the end of the study (blue squares), when the care units had a small number of COVID-19 patients hospitalized. The subjects’ mean PWV at the beginning of the study was 9.12 m/s, with a standard error of mean 0.31 m/s, and at the end of the study, the subjects’ mean PWV was 8.64 m/s, with the standard error of the mean of 0.27 m/s. Therefore, when the care unit had many hospitalized COVID-19 patients, the medical care workers’ mean PWV was higher than the mean PWV at the end of the study, when the number of patients with COVID-19 hospitalized was small.

Additionally, by comparison with the European studies [31], one can observe that the mean PWV in our studied group (see Table 5) is higher than the mean PWV for healthy people in Europe for the same age group. Namely, according to The Reference Values for Arterial Stiffness’ Collaboration [31], in Europe’s healthy population, the average PWV is 6.5 m/s in the age group 20–40 and 8.3 m/s in the age group 40–60.

Figure 3 shows the dependence of PWV on the stress questionnaire score of subjects participating in the study, both at the beginning and at the end of the study, and the linear regression lines are shown.

Pearson’s correlation coefficient, r, as well as the linear fit parameters, are presented in Table 6. The positive value of r is in agreement with the increasing slope of the linear regression line, but the value of r shows that there is a weak correlation between PWV and the stress score. An inconsistent correlation between the level of stress at work and PWV has also been reported by Nomura et al. [19] and Kantermann et al. [20].

On the other hand, there are various studies demonstrating the correlation between acute mental stress and PWV [32,33,34,35,36,37]. We must mention that the mental stress defined in these studies refers to mathematical operations or mental activities performed under time pressure and mistakes were immediately mentioned, so that the subjects were under time pressure and also embarrassed about making mistakes. At the end of the test, the subjects were asked to rate negative emotions, and a total score was correlated to the pulse wave velocity (PWV) [32,33,34,35,36,37].

A recent study [38] analyzed the associations between arterial stiffness and life stressors (such as job strain, sleep loss due to worry, illnesses or death in the family, financial difficulties and caregiving) that can lead to increased PWV among late middle-aged persons from Finland. In this study, the authors observed increased PWV among the subjects who reported recent illness or death in the family or financial difficulties within the past 12 months [38]. Life stressors such as job strain, sleep loss due to worry and caregiving were not associated with increased PWV [38].

As is known, each person is exposed both to conscious stress and unconscious stress [39]. Acute mental stress at work can be forgotten in the calm, pleasant atmosphere of the family, so in many cases, stress at work can become unconscious stress and cannot be identified by general stress questionnaires, especially among people with long experience at work. Therefore, the changes in PWV presented in Figure 2 may be explained by the change in unconscious stress, and this is the reason for the lack of a strong correlation between the PWV values and the stress score resulting from the questionnaire.

## 5. Conclusions

According to the results of the stress questionnaire, the health care workers had medium to high levels of stress that were higher during the period with a high number of COVID-19 patients than during the period with a low number of COVID-19 patients.

The average PWV of health care workers during the period with a high number of COVID-19 patients was higher than their average PWV during the period with a low number of COVID-19 patients. Because the only change in the life of the health care workers during the study was related to their work, the decrease in their PWV is related to the decrease in stress at work.

Pearson’s correlation coefficients show a weak correlation between the stress score and PWV, both at the beginning and at the end of the study.

We see a necessity to address the recognition of occupational stress and implement measures to reduce it to the lowest possible level to prolong the health and well-being of medical workers.

## Figures and Tables

**Figure 1 medicina-58-00704-f001:**
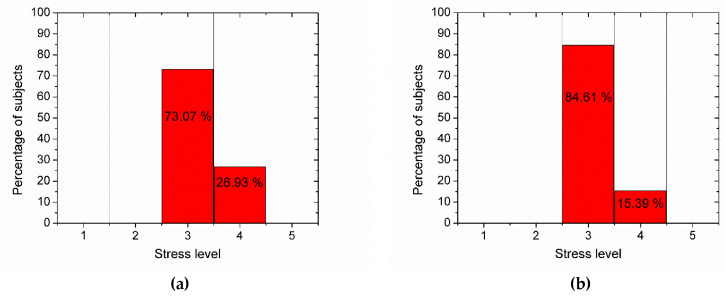
Evaluation of the stress level in the studied group (**a**) at the beginning and (**b**) at the end of the study (level 1, very low: ≤48 points; level 2, low: 49–72 points; level 3, normal: 73–120 points; level 4, high: 121–144 points; level 5, dangerously high: ≥145 points).

**Figure 2 medicina-58-00704-f002:**
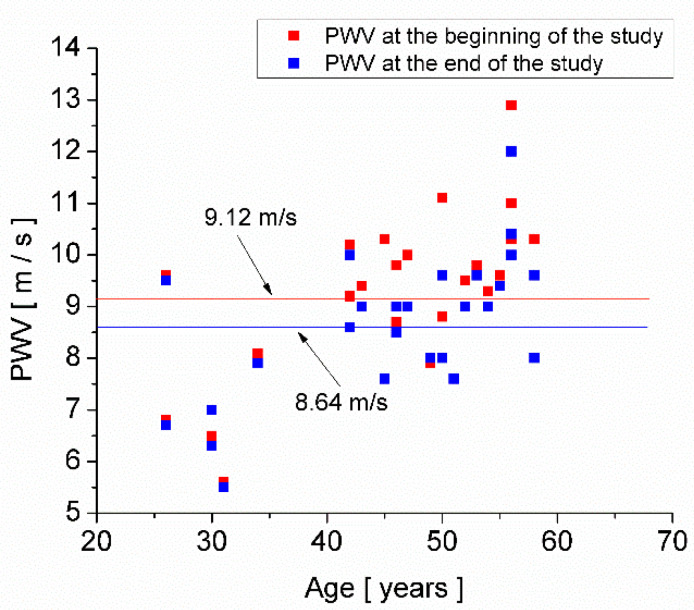
Pulse wave velocity (PWV) versus age of each subject at the beginning of the study (red squares) and at the end of the study (blue squares).

**Figure 3 medicina-58-00704-f003:**
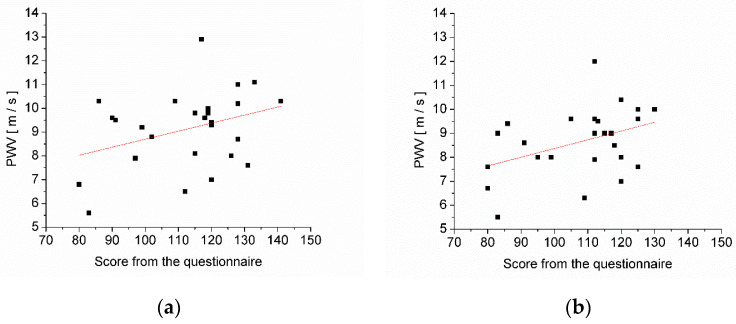
The plot of PWV versus the stress questionnaire score and the linear regression line: (**a**) at the beginning of the study and (**b**) at the end of the study.

**Table 1 medicina-58-00704-t001:** Characteristics of the subjects in the study group.

Parameter	Mean Value	Standard Deviation
Age	45.61 years	10.20 years
Seniority at work	22.34 years	12.65 years
Height	1.65 m	0.074 m
Weight	72.57 kg	15.93 kg
BMI	26.45 kg/m^2^	4.49 kg/m^2^
Systolic BP	115.96 mmHg	12.05 mmHg
Diastolic BP	69.07 mmHg	9.19 mmHg
Heart rate	72.03 bpm	8.82 bpm

**Table 2 medicina-58-00704-t002:** The results of the paired-sample *t*-test for the stress questionnaire score.

	Mean [m/s]	SD [m/s]	SEM [m/s]
Stress score at the beginning of the study	112.57	16.65	3.26
Stress score at the end of the study	107.65	15.44	3.02
Null hypothesis	Stress score at the beginning of the study–Stress score at the end of the study = 0		
Alternate hypothesis	Stress score at the beginning of the study is different from the stress score at the end of the study		
*p* (from *t*-test) ^1^		7.57∙10^−9^	
Alpha		0.05	
Power		1	

^1^ Below the 0.05 level, the difference of the means is statistically significant.

**Table 3 medicina-58-00704-t003:** The results of the stress questionnaire by age group.

Age Group	Average Score at the Beginning of the Study	SD at the Beginning of the Study	Average Score at the End of the Study	SD at the End of the Study
20–40	104.66	18.17	102.83	16.94
41–60	114.95	15.88	109.10	15.13

**Table 4 medicina-58-00704-t004:** The results of the paired-sample *t*-test for the mean values of PWV from Figure 2.

	Mean [m/s]	SD [m/s]	SEM [m/s]
PWV at the beginning of the study	9.12	1.61	0.31
PWV at the end of the study	8.64	1.39	0.27
Null hypothesis	PWV at the beginning of the study–PWV at the end of the study = 0		
Alternate hypothesis	PWV at the beginning of the study is different from PWV at the end of the study		
*p* (from *t*-test) ^1^		3.32∙10^−4^	
Alpha		0.05	
Power		0.97	

^1^ Below the 0.05 level, the difference of the means is statistically significant.

**Table 5 medicina-58-00704-t005:** The PWV results by age group.

Age Group	PWV at the Beginning of the Study [m/s]	SD at the Beginning of the Study [m/s]	PWV at the End of the Study [m/s]	SD at the End of the Study [m/s]
20–40	7.26	1.40	7.15	1.39
41–60	9.68	1.81	9.09	1.06

**Table 6 medicina-58-00704-t006:** The correlation and linear fit results.

Dependence of PWV on the Stress Questionnaire Score	Slope	Intercept	Pearson’s Correlation Coefficient	Significance ^2^
at the beginning of the study	0.033	5.325	0.348	0.080
at the end of the study	0.036	4.728	0.403	0.041

^2^ for correlation, a 2-tailed test of significance was used.

## Data Availability

The study protocol and supplementary information will be provided upon request.

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
