# Peer review of "Pulse Wave Velocity, a Predictor of Major Adverse Cardiovascular Events, and Its Correlation with the General Stress Level of Health Care Workers during the COVID-19 Pandemic"

_medicina, 2022, doi:10.3390/medicina58060704_

Round 1

Reviewer 1 Report

interesting work topical. All the obtained results were discussed in the discussion In conclusion, the correlation between stress score and PWV at the end of the study should be mentioned

Reviewer 2 Report

The article covers an interesting subject, about the physiological impact of the stressful environment of Covid19 hospitals on the medical personnel.

There are a few drawbacks:

  • The Introduction is too large and difficult to follow.
  • The number of study subjects is too low.
  • The statistics is very briefly described in the Methods section. A multivariate analysis is lacking.

Reviewer 3 Report

This is an interesting study regarding stress of medical staff under COVID-19.

However, I have some questions to be clarified by authors.

Authors documented that employees in healthcare departments from Timiosoara were recruited.

Does it mean a specific single center in Timiosoara? Or, did they recruit medical staffs in several hospitals?

I also ask authors to clarify what is healthcare department. Author list included several departments and I assume staffs from many departments were included. It is important for authors to describe from where departments staffs were recruited. Whether staffs were recruited from respiratory medicine or any other departments was important, because staffs who don’t care patients with respiratory problem would tend to feel more stress than staffs from the respiratory medicine. I recommend authors compare stress level in the point of difference of department.

Authors explained one of the medical professions as “maids”. Does it mean nursing assistant or cleaning staff? I feel that it is strange to use the word for people working in hospital.

Authors explained that there were “many” and “few” COVID-19 patients in the beginning and end of the study, respectively. Authors should provide objective data instead of subjective expression such as many or few.
